# Using whole blood cultures in interferon gamma release assays to detect *Mycobacterium tuberculosis* complex infection in Asian elephants (*Elephas maximus*)

Chitsuda Pongma[1,2], Songkiat Songthammanuphap[3], Songchan Puthong[4], Anumart Buakeaw[4], Therdsak Prammananan[5], Saradee Warit[5], Wanlaya Tipkantha[6], Erngsiri Kaewkhunjob[6], Waleemas Jairak[6], Piyaporn Kongmakee[6], Choenkwan Pabutta[7¤], Supaphen Sripiboon[8], Wandee Yindeeyoungyeon[5☯*], Tanapat Palaga[2,3☯*]

1 Graduate Program in Biotechnology, Faculty of Science, Chulalongkorn University, Bangkok, Thailand, 2 Center of Excellence in Immunology and Immune-Mediated Diseases, Chulalongkorn University, Bangkok, Thailand, 3 Department of Microbiology, Faculty of Science, Chulalongkorn University, Bangkok, Thailand, 4 Institute of Biotechnology and Genetic Engineering, Chulalongkorn University, Bangkok, Thailand, 5 The National Center for Genetic Engineering and Biotechnology, National Science and Technology Development Agency (NSTDA), Pathum Thani, Thailand, 6 Bureau of Conservation and Research, Zoological Park Organization of Thailand, Bangkok, Thailand, 7 Elephant Kingdom Project, Zoological Park Organization of Thailand, Surin, Thailand, 8 Department of Large Animals and Wildlife Clinical Science, Faculty of Veterinary Medicine, Kasetsart University, Nakhon Pathom, Thailand

☯ These authors contributed equally to this work.
¤ Current address: The Monitoring and Surveillance Center for Zoonotic Diseases in Wildlife and Exotic Animals, Faculty of Veterinary Science, Mahidol University, Nakhon Pathom, Thailand
* tanapat.p@chula.ac.th (TP); wyindeeuga@gmail.com (WY)

## Abstract

Elephants are susceptible to *Mycobacterium tuberculosis* (M. tb) complex (MTBC) infections. Diagnosis of tuberculosis (TB) in elephants is difficult, and most approaches used for human TB diagnosis are not applicable. An interferon gamma release assay (IGRA) to diagnose TB in Asian elephants (*Elephas maximus*) using peripheral blood mononuclear cells (PBMCs) has been previously developed. Although the assay is shown to be valid in determining MTBC infection status, the laborious PBMC isolation process makes it difficult to use. In this study, we simplified the method by using whole blood cultures (WC) as the starting material. Using PBMC cultures for IGRA, the MTBC infection status of 15 elephants was first confirmed. Among these animals, one has been previously confirmed for M. tb infection by both TB culture and PCR and the other was confirmed for MTBC infection in this study by droplet digital PCR (ddPCR) method. WC for IGRA consisted of an unstimulated sample, a mitogen stimulated sample, and sample stimulated with recombinant M. tb antigens, ESAT6 and CFP10. Using WC for IGRA in the 15 enrolled elephants, the results showed that 7 out of 15 samples yielded MTBC infection positive status that were completely concordant with those from the results using PBMCs. To test this method, WC for IGRA were applied in another elephant cohort of 9 elephants. The results from this cohort revealed a perfect match between the results from PBMC and WC. Responses to ESAT6 or CFP10 by PBMC and WC were not completely concordant, arguing for the use of at least two M. tb antigens

**Data Availability Statement:** All relevant data are within the paper and its Supporting information files.

**Funding:** This research was supported by the National Science and Technology Development Agency (grant number NSTDA: Project ID P-18-52114) awarded to WY. CP is supported by a Science Achievement Scholarship of Thailand (SAST) and the 90th Anniversary of Chulalongkorn University Fund (Ratchadaphiseksomphot Endowment Fund). TP is supported by the National Research Council of Thailand (grant number 811/2563).

**Competing interests:** The authors have declared that no competing interests exist.

for stimulation. Given the ease of sample handling, smaller blood sample volumes and equivalent efficacy relative to the PBMC approach, using WC for IGRA provides a novel, rapid, and user-friendly TB diagnostic method for determining the MTBC infection in elephants.

## Introduction

Tuberculosis (TB) is a potentially serious infectious disease that affects humans and animals. It is caused by the *Mycobacterium tuberculosis* complex (MTBC), especially *Mycobacterium tuberculosis* (M. tb), and spreads from person to person by airborne transmission [1]. In addition to human-to-human transmission, M. tb can be transmitted from animals to humans and *vice versa*. The main cause of TB infections in animals consists of transmission by close prolonged contact with individuals or animals with active TB [2]. MTBC infects a wide variety of animal hosts, including nonhuman primates and captive and wild animals, such as cattle, rhinoceros, wild dogs, and elephants [1, 3, 4]. In various countries, including Thailand, captive Asian elephants (*Elephas maximus*), in particular, often have prolonged periods of close contact with humans and are prone to infection with M. tb, as first reported in 1875 [2]. There is also a report of fatal TB in a free-ranging African elephant in a region with a high human TB burden [5]. TB in elephants presents as a chronic infection and often does not exhibit obvious clinical symptoms until the disease reaches the active stage [6]. The symptoms in infected elephants include fatigue, labored breathing, trunk discharges, and weight loss, followed by death [7].

Owing to their large size and thick skin, diagnostic tests for TB in elephants are difficult to perform, and the regular techniques used in humans, such as chest X-rays and tuberculin skin tests, are not applicable [8]. As a result, an accurate and user-friendly diagnostic platform for elephant TB that covers all stages of infection is urgently needed. Currently, bacterial cultures from trunk washes and affected tissues at necropsy are recommended as the current gold standard methods for TB diagnoses. However, such methods have various limitations, including difficulty in obtaining samples, low sensitivity, and the limitation of detection only in the active TB stage. As an alternative approach, other immunological indicators of TB infections, such as antibody detection and antigen-specific interferon release assays, have been investigated [8–12].

M. tb is an intracellular pathogen, and cell-mediated immune responses (CMIR) are elicited early after infection and are considered to mainly contribute to the control of infection [13]. Interferon gamma (IFN©) is a key cytokine that is produced mainly by T helper type 1, cytotoxic T lymphocytes, and natural killer cells during mycobacterial infections [14]. A test to detect memory T cells that are specific to TB antigens can be implemented by using an IFN© release assay (IGRA). This method uses defined TB antigen stimulation of lymphocytes, followed by measurements of IFNγ release using an enzyme-linked immunosorbent assay (ELISA). IGRA is currently used for human TB diagnoses, which uses whole blood samples for stimulation by antigens within the region of difference 1 (RD1) of M. tb. These antigens are peptides derived from early secreted antigenic target 6 kDa (ESAT6) and culture filtrate protein 10 kDa (CFP10), which are not present in the vaccine strain, Bacillus Calmette-Guérin (BCG). These antigens are potent inducers of CMIR, even in an *in vitro* setting, which enables detection of IFN© secretion [15, 16].

Previous studies have demonstrated that IGRAs can be used as an alternative tool to diagnose TB in Asian and African elephants [12, 17, 18]. In our previous report, we developed an

in-house IGRA to detect M. tb infections that can distinguish between naïve, nontuberculous mycobacteria (NTM) infections and MTBC infections in Asian elephants with the world's largest sample sizes of more than 60 elephants [12]. The protocol is relatively accurate in detecting MTBC infections, but there are some disadvantages in using peripheral blood mononuclear cells (PBMCs) for measuring the response to stimulating antigens. Although using PBMCs helps in normalizing the discrepancies in white blood cell counts in samples, isolating PBMCs is time-consuming and requires a tissue culture facility with skilled technicians [12].

Therefore, the aim of this study was to develop an IGRA-based TB diagnostic platform for use in elephants using whole blood cultures. The results obtained from this study will lead to a user-friendly IGRA-based elephant TB diagnosis platform.

## Materials and methods

### Blood samples from Asian elephants

Blood samples (n = 22) were collected from captive Asian elephants (*E. maximus*) in Thailand (20 females and 2 males, aged 3–80 years). The elephants were registered under the health checkup program by the Zoological Park Organization and National Elephant Institute. Whole blood samples were collected periodically in heparinized tubes (BD Vacutainer, Becton Dickinson, USA) and processed within 24 hr.

For testing IGRA in a cohort with unknown M. tb infection status, blood samples (n = 9) were collected from another captive elephant cohort (female n = 7 male n = 2, aged 6–48 years). Blood samples were collected in heparinized tubes (BD Vacutainer, Becton Dickinson, USA) and processed within 24 hr. Blood samples were subjected for peripheral blood mononuclear cells (PBMCs) isolation or used as whole blood for IGRA as described below.

All procedures for collecting blood samples were approved by the Institutional Animal Care and Use Committee (IACUC) of the Zoological Park Organization of Thailand and the National Center for Genetic Engineering and Biotechnology (BT-Animal 1/2562) and were performed according to the IACUC guidelines. Detailed information on elephants enrolled in this study is summarized in S1–S3 Tables.

### Isolation of PBMCs

PBMCs were isolated by using the density centrifugation method with Ficoll-Paque Premium media solution (GE Healthcare, USA) as previously described [12]. Briefly, blood samples were centrifuged at 400 x g to obtain leukocytes (buffy coat). The separated leukocytes were diluted with phosphate-buffered saline (PBS) to adjust the volume to 10 ml to resuspend the cell pellets. After dilution, the diluted cell suspensions (10 ml) were carefully overlaid onto 5 ml of Ficoll-Paque Premium media solution (GE Healthcare) in 15-ml centrifuge tubes, and the samples were centrifuged at 400 x g for 40 min at 20°C without breaking. The mononuclear cells that formed a layer between the plasma (upper layer) and Ficoll-Paque Premium media were collected into new centrifuge tubes and washed. The washing step was repeated three times before resuspending the cell pellets in complete RPMI 1640 media supplemented with 0.05 mM β-mercaptoethanol.

### Reagents

Recombinant ESAT6 and CFP10 were produced by using *Escherichia coli* and purified by affinity column chromatography as previously reported [19]. Recombinant eIFN© was produced in *Escherichia coli* transformed with an expression plasmid harboring the eIFN© gene

obtained from a female Asian elephant. Monoclonal antibodies to detect eIFN© were prepared from hybridomas obtained by immunizing mice with recombinant eIFN© [12].

## Stimulating PBMCs with mitogens or antigens

PBMCs were seeded at $1 \times 10^5$ cells/well in a 96-well U-bottom plate (Thermo Fisher Scientific, USA) in a final volume of 150 μl. Cells were stimulated with concanavalin A (ConA) (10 μg/ml) (Sigma Aldrich, USA), recombinant ESAT6 and/or CFP10 (10 μg/ml each) for 72 h in a $CO_2$ incubator. After stimulation, culture supernatants were collected and subjected to sandwich ELISA to measure eIFN© as described below.

## Whole blood culture for IGRA

Blood samples were collected from each elephant into heparinized tubes (BD Vacutainer). One milliliter of blood from each heparinized tube was transferred into 5-ml polypropylene tubes (Falcon A Corning Brand, USA). Mitogens (pokeweed mitogen (PWM) (10 μg/ml), ConA (10 μg/ml), PWM with ConA (10 μg/ml each)) or specific TB antigens (recombinant ESAT6 (20 μg/ml) and CFP10 (20 μg/ml)) were added at the indicated concentrations to stimulate immune cells in whole blood. The tubes were gently mixed by shaking for 10 times, and incubation was carried out at 37˚C for 24 hr. After 24 hr of stimulation, the blood tubes were centrifuged, and plasma samples were harvested to detect eIFN© by sandwich ELISA as described for the PBMC cultures.

## Sandwich ELISA to detect eIFNγ

All reagents used for sandwich ELISA were prepared according to previously described procedures [12]. The 96-well ELISA plates were coated with goat anti-rabbit IgG Fc fragment-specific antibody (Jackson ImmunoResearch, USA). After blocking with 10% FBS in PBS, the plates were washed, and diluted rabbit sera specific to eIFNγ were added to the wells. This step was followed by adding standard eIFNγ (0–10 ng/ml), samples from the PBMC culture supernatants or plasma from whole blood cultures to the plates. Plates were kept overnight at 4˚C. Plates were washed the following day before a mouse monoclonal IgG antibody specific to eIFNγ (0.5 μg/ml) was added. The plates were incubated at 37˚C for 1 hr, and goat anti-mouse IgG (Sigma–Aldrich) conjugated with peroxidase (1:10000) was added, followed by TMB substrate solution. The reaction was stopped with 1 M $H_2SO_4$, and the absorbances at 450 nm were measured by a microplate reader (Multiskan FC, Thermo Fisher Scientific, USA).

The limit of detection (LOD) for the standard curve was calculated using the following formula: LOD = $B_0$+3SD, where $B_0$ is the average absorbance at 450 nm of the blank samples and SD is the standard deviation of the blank samples. Specific M. tb antigens stimulated -samples that yielded eIFNγ higher than $B_0$+3SD and greater than the lowest concentration of the standard curve were considered positive results.

## Results

### Status of MTBC infections in the enrolled elephants using PBMC IGRA

To confirm the MTBC infection status in the cohort of elephants enrolled in this study, we applied IGRA using isolated PBMCs as described previously [12]. Among these elephants, elephant No. 2 has been previously diagnosed as M. tb infection by both TB culture and PCR from trunk wash [20] and No.1 has been confirmed of MTBC infection by ddPCR from trunk swab (this study, S1 File). Freshly isolated PBMCs were cultured in the presence of purified recombinant ESAT6 or CFP10 or mitogen ConA for 72 hr. The culture supernatants were

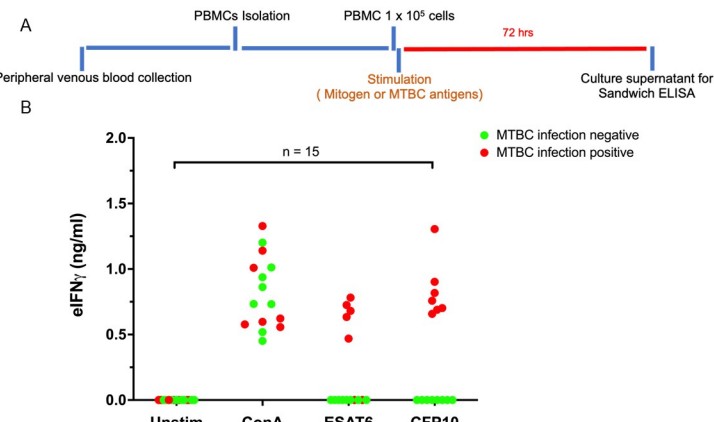

**Fig 1. eIFNγ amounts released after stimulation of elephant PBMCs by mitogens and antigens.** (A) Schematic representation of PBMC stimulation for IGRA. (B) The eIFN© levels in culture supernatants of PBMCs stimulated by MTBC antigens as indicated in (A). The eIFN© levels were measured by sandwich ELISA. Different color-coded dots represent individual elephants with MTBC infection statuses determined by using the criteria described in previous reports [12].

harvested and subjected to ELISA to determine the eIFNγ concentrations (Fig 1A). The eIFNγ concentrations obtained from each elephant are summarized in Fig 1B. Based on the interpretation criteria described by Songthammanuphap *et al.* with some modifications, the MTBC infection status was divided into 2 groups: 1) MTBC infection negative, 2) MTBC infection positive. The infection status is depicted as color-coded dots in Fig 1B. The results of the infection status of the 15 elephants are summarized and presented as a heatmap in Fig 2.

Among the 15 elephants tested, 8 elephants were negative for MTBC infection whereas 7 elephants showed MTBC positive results. Among the ones tested positive for MTBC infection, elephant No. 1 was diagnosed with M. tb infection by ddPCR in this study and elephant No. 2 has been previously diagnosed with M. tb infection by both PCR and TB culture. Interestingly, sample No. 2 and 3 showed negative response to ESAT6 stimulation but exhibited a positive response to CFP10 and, thus, considered MTBC positive.

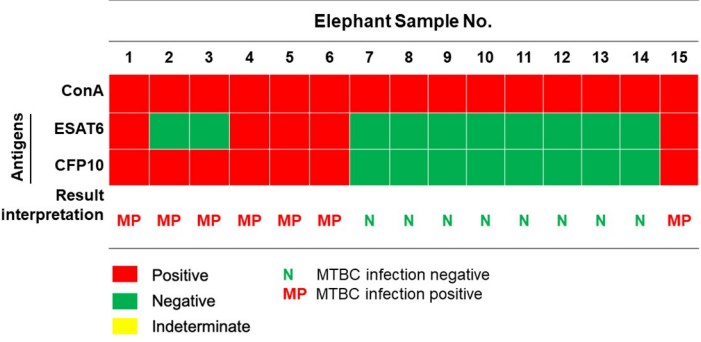

**Fig 2. Summary of MTBC infection status determined by PBMC cultures for IGRA.** The responses of PBMCs against M. tb antigens by eIFN© secretion were used to determine MTBC infection status. The results are summarized in a heatmap, where an MTBC positive status is shown by red boxes, MTBC infection negative status is shown by green boxes, and infection indeterminate status is shown by yellow boxes.

### Testing mitogens for whole blood culture and detection of eIFNγ

In our previous elephant PBMC culture study, ConA and PWM were tested for their ability to stimulate secretion of eIFN©, and the results showed that ConA was superior to PWM [12]. Therefore, ConA was chosen as a mitogen as the positive control for PBMC stimulation. To select a suitable mitogen for whole blood cultures, blood samples from 10 healthy elephants were subjected to mitogen stimulation, and the eIFN© concentrations were determined. As shown in Fig 3A, PWM (10 μg/ml) alone or PWM in combination with ConA (10 μg/ml each) induced significantly higher eIFN© secretion levels than the unstimulated or ConA-treated samples. To our surprise, ConA alone did not induce detecstable amounts of eIFN©, in contrast to those in PBMC cultures. Therefore, in whole blood cultures for IGRA, PWM was chosen as a positive control to stimulate eIFN© secretion.

### Protocol for whole blood cultures for IGRA and interpretation of the results

To establish the whole blood culture for IGRA, the overall protocol is summarized in Fig 3B. Venous blood was collected in heparinized tubes with at least 4 ml per animal and used within 24 hr of collection. Blood was aliquoted into 4 polypropylene tubes (1 ml each). A negative control tube was left unstimulated (NIL), while PWM (10 μg/ml) was added to the mitogen positive control tube. For the specific TB antigens, recombinant ESAT6 (20 μg/ml) was added to the TB1 tube, and recombinant CFP10 (20 μg/ml) was added to the TB2 tube. After thoroughly mixing the samples by gentle shaking, the tubes were incubated at 37°C for 24 hr. Plasma samples were harvested by centrifugation and used as samples for ELISA to detect eIFN©. The results obtained from ELISA were calculated to obtain the eIFN© concentrations. When the absorbance at 450 nm from ELISA fell between the LOD and lowest concentration of the standard curve, the samples were considered indeterminate.

The flowchart in Fig 4 depicts the interpretation of the eIFN©secretion results and MTBC infection status. First, the value of the mitogen-stimulated positive control must be higher than that of the negative control. If this is not the case, the result is determined to be uninterpretable. For the samples that show the expected results of the positive control tubes, the results from the samples with specific TB antigen stimulation will be evaluated. If both TB1 and TB2

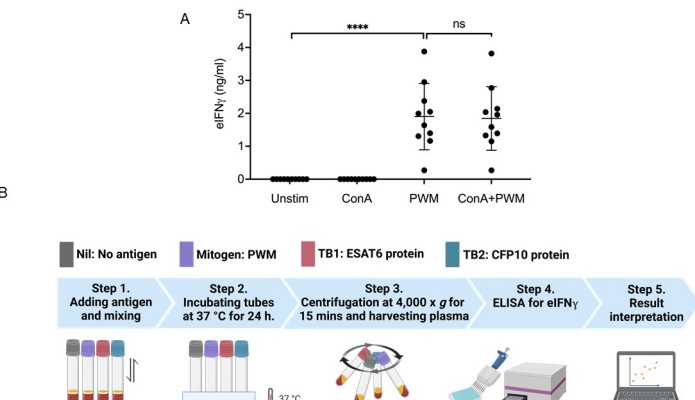

**Fig 3. Schematic representation of whole blood cultures for IGRA.** (A) Amounts of eIFNγ produced from cells in whole blood cultures after stimulation with different mitogens for 24 h (n = 10). **** *p value* ≤ 0.0001, and ns indicates not statistically significant with a *p value* > 0.05. (B) Schematic workflow for whole blood cultures and IGRA for MTBC detection. The figure is created by the authors with BioRender.com.

yield results similar to the negative control, the infection status will be determined as MTBC infection status negative. If at least one of the TB1 or TB2 tubes (ESAT6 and CFP10) shows a response higher than that of the negative control, the status will be interpreted as MTBC infection positive. If at both TB1 and TB2 yields an indeterminate readout, the status will be marked as indeterminate.

## Whole blood cultures for IGRA to determine MTBC infection status

Using the method and interpretation described above, the 15 enrolled elephants with determined MTBC infection statuses were subjected to whole blood cultures for IGRAs. The eIFN© concentrations obtained from PBMC cultures and whole blood cultures were compared and the results are shown in Fig 5A. The color-coded dots represented the MTBC infection status interpreted by the criteria described by Songthammanuphap *et al.* [12] for PBMC or those described above for WC. All but two whole blood culture samples yielded higher eIFN© concentrations than the PBMC cultures in the mitogen-stimulating positive control. Similarly, in ESAT6- and CFP10-stimulated conditions, all but two whole blood culture samples showed higher eIFN© concentrations than the PBMC cultures. In the TB1 (ESAT6) tubes, five samples (No. 2, 3, 5, 6, 15) yielded conflicting results between the whole blood cultures and PBMC cultures. In the TB2 (CFP10) tubes, all samples yielded consistent results between PBMC and WC cultures.

Based on the IGRA interpretation described in Fig 4, the results of the two IGRA approaches are summarized as a heatmap in Fig 5B. Among the tested elephants, 7 elephants (e.g., No. 1, 2, 3, 4, 5, 6 and 15) yielded clear MTBC infection positivity in the whole blood culture assays, consistent with the PBMCs for IGRA results (46.7% of total). In contrast, 8 elephants (e.g., No. 4, 8, and 10–15) were negative for the MTBC infection status (53.3% of total) based on the whole blood culture assays. All samples that yielded negative infection status showed consistent response to both TB1 and TB2 antigens. In contrast, elephants that showed MTBC infection-positive results exhibited contradictory responses in PBMC and WC cultures to ESAT6.

Next, to test the WC culture for IGRA in an elephant cohort with unknown MTBC infection status, we obtained blood samples from additional 9 elephants and the PBMC and WC cultures were performed in parallel. The concentrations of eIFN© obtained by the two approaches are shown in Fig 6A and the interpretation of the results are summarized in Fig 6B. Similar to the results from previous cohort, WC yielded higher concentrations of eIFN© than PBMC culture. Furthermore, all samples that yielded negative infection status have

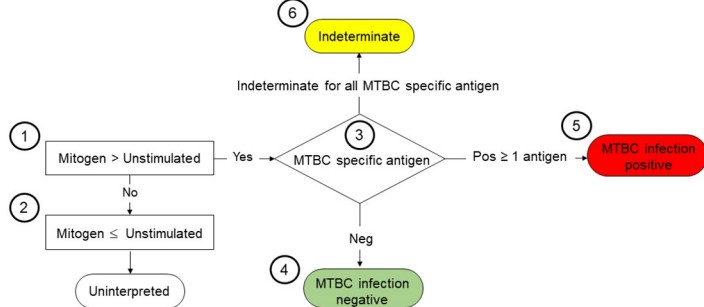

**Fig 4. Flowchart detailing the interpretation of results from whole blood culture IGRAs.** Based on the IGRA results, the interpretation of infection status is conducted by following steps 1–6. The interpretations results include uninterpreted, MTBC infection negative, MTBC infection positive and MTBC infection indeterminate.

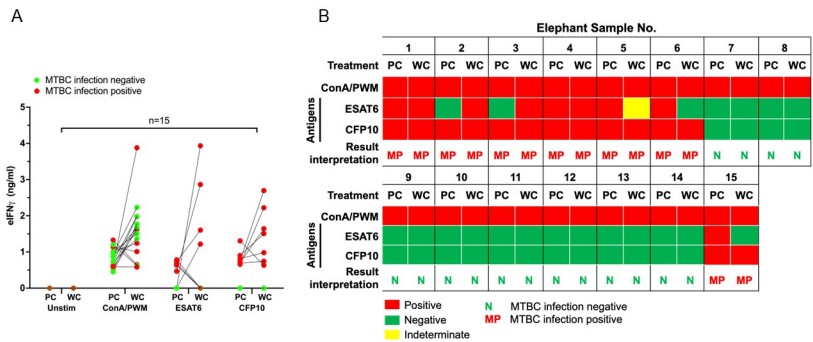

**Fig 5. Results of eIFNγ concentrations obtained by PBMC cultures IGRA (PC) and whole blood cultures IGRA (WC) and summary of result interpretations.** (A) The results are presented as the eIFN© concentrations with comparisons between PBMC assays or whole blood culture assays at the individual level. The MTBC infection statuses were simplified to infection negative (green) or infection positive (red). (B) Fifteen elephant samples were subjected to PBMC cultures for IGRA in parallel with whole blood cultures for IGRA. The results obtained from each assay were interpreted based on the criteria described by Songthammaniphap *et al*. and above [12]. The interpretations are presented as a heatmap.

consistent negative responses in both WC and PBMC cultures (Fig 6A and 6B). In contrast, samples that were interpreted as MTBC positive exhibited conflicting responses to at least one M. tb antigens (No. 21, 22, 23, 24). These results strongly argue for the use of at least 2 M. tb antigens for IGRA. Taken together, whole blood cultures for IGRA provide a user-friendly platform with less amount of blood samples required to determine the MTBC infection status in elephants.

## Discussion

TB infections in captive and wild elephants with severe outcomes have been increasingly reported and can be fatal [5, 21, 22]. Diagnosis of TB in elephants is a key step toward controlling the spread of this pathogen. Diagnostic tests not only help in veterinary care for the infected animals but also reduce the risk of human exposure to the pathogen because elephants sometimes live in close contact with humans in some parts of the world, including Thailand

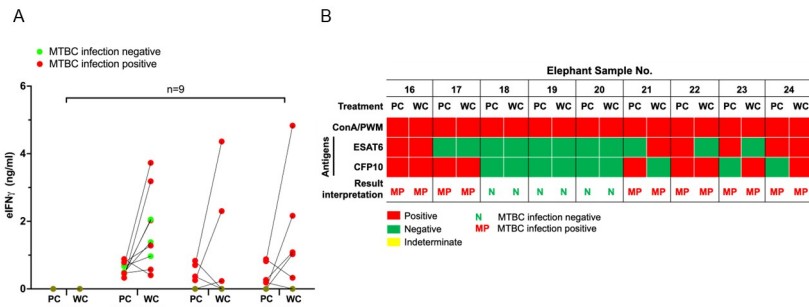

**Fig 6. The eIFNγ concentrations obtained by PBMC cultures IGRA (PC) and whole blood cultures IGRA (WC) and summary of result interpretations in elephant cohort with unknown MTBC infection status.** (A) The results are presented as the eIFN© concentrations with comparisons between PBMC assays or whole blood culture assays at the individual level. The MTBC infection statuses were simplified to infection negative (green) or infection positive (red). (B) Nine elephant samples were subjected to PBMC cultures for IGRA in parallel with whole blood cultures for IGRA. The results obtained from each assay were interpreted based on the criteria described by Songthammaniphap *et al*. and above [12]. The interpretations are presented as a heatmap.

[20]. The gold standard for elephant TB diagnosis by using trunk wash cultures has many limitations, including difficulty in collecting samples and low sensitivity. Alternative methods for detection of active M. tb infection such as quantitative PCR (qPCR) or a highly sensitive droplet digital PCR (ddPCR) techniques have advantages over classical procedure for elephant TB diagnosis [23]. As in human TB diagnoses, IGRA presents an alternative method for diagnosing tuberculosis in elephants. Previously, IGRAs for elephant TB diagnosis have been reported and were demonstrated to be able to detect MTBC infections both in the latent and active disease stages [12, 17, 18].

In our previous study, we developed an IGRA to detect MTBC and NTM infections in Asian elephants in captivity who live in close proximity to humans in Thailand. The number of elephants subjected to IGRAs for TB diagnosis is the largest in the world to date. Tuberculosis is known to be transmitted from humans to animals, and elephant TB is shown to be transmitted back to humans in close contact [24]. IGRAs have widely been used for TB diagnosis in humans because of their sensitivity and ability to distinguish between BCG-vaccinated and TB-infected individuals. For animal TB, IGRA has been experimentally performed on wild and captive animals, but none are in routine use or commercially available [22]. The use of PBMCs in IGRAs has an advantage in normalizing the discrepancies in white blood cell numbers among individual elephants. The assay, however, requires laborious procedure of isolation of PBMCs and skilled laboratory personnel to work with relatively large volumes of blood. This study found that both PBMCs and whole blood cultures yielded similar interpretation of MTBC infection status and proposed the use of whole blood culture instead of PBMCs for IGRA in elephants.

Overall, stimulation of whole blood cultures by MTBC antigens yielded similar or higher levels of eIFN© production compared with those of PBMC cultures for IGRA. However, each elephant might respond to ESAT6 and CFP10 stimulation in PBMCs and whole blood cultures differently. For ESAT6 stimulation, 7 out of 24 samples showed discrepancies between PBMC and whole blood cultures. For CFP10, 3 out of 24 samples yielded contradictory results between the two approaches. CFP10 stimulation showed more consistent responses in MTBC infection positive elephants over ESAT6. Therefore, the use of at least two M. tb antigens are needed to obtain accurate interpretation of the infection status. In addition, combining the two antigens in one sample tube or using ESAT6-CFP10 fusion protein may provide a better approach for use in IGRAs using whole blood cultures [25].

On the attempt to simplify the IGRA method for elephant, we employed whole blood culture instead of PBMCs culture. In addition, omitting PPDA and PPDB used in previous study [12], and used only the defined recombinant proteins (e.g., ESAT6 and CFP10) to stimulate the recall response [26] in this streamline assay, make it possible to use blood volumes as small as 4 ml for each diagnostic test, in contrast to the 20 ml or greater volumes required for the PBMC isolation approach. Furthermore, the assay duration was shortened to 24 hr, as opposed to 72 hr with PBMCs.

Interestingly, suitable mitogens were tested as a positive control, and PWM was found to induce robust eIFN© production from whole blood cultures. This result is in line with a previous study where PWM and phorbol myristate acetate plus ionomycin induced high eIFN© production from whole blood cultures [17]. In a PBMC assay, ConA and PWM were found to induce similar levels of eIFN© production [12]. PWM has been known and used to stimulate T and B cells in humans and mice with mechanisms that are not well described [27]. Monocyte-depleted PBMCs are unresponsive to PWM stimulation, suggesting an important role of monocytes/macrophages [28]. ConA failed to stimulate whole blood cells to produce eIFN©even though it can robustly stimulate PBMCs. We speculate that the stimulation durations (e.g.,

24 hr for whole blood culture vs. 72 hr for PBMC culture) may be responsible for the difference in eIFN© quantities.

Interpretation of the outcomes of IGRAs is also simplified to only MTBC infection negative or positive. Since crude antigens, PPDA and PPDB, representing NTM antigens and MTBC antigens, differentiation between MTBC and NTM infections may be possible if PPDA and PPDB are incorporated. However, NTM infections in theory are considered MTBC infection negative which usually have no clinical significance.

In the current study, WC for IGRA was first evaluated in an elephant cohort of 15 enrolled animals with previously known MTBC infection status using PBMC culture for IGRA. Because the gold standard for active elephant TB diagnosis is the trunk wash culture, we added elephant No.2 that has been previously diagnosed with TB infection by positive for both TB culture and PCR from trunk wash, and elephant No.1 that has been shown to have M. tb in trunk swab by ddPCR, as "true" positive control ([20] and S1 File). In addition, elephant No. 24 (cohort of 9 elephants) is also considered a "true positive of TB infection", because it has been confirmed to have M. tb in trunk swab by ddPCR in this study (S1 File). The test for latent TB infection in elephants by IGRA have not been documented but the use of humoral immune indicator by detecting TB antigen specific immunoglobulins have been proposed [8]. The use of antibody detection may have advantage in monitoring response to treatment, but the level of antibody is dynamic and may decrease overtime after exposure to M. tb.

In conclusion, this study describes a novel platform that can diagnose MTBC infections using whole blood cultures. This approach has several advantages over PBMC for IGRA, including ease of sample preparation (user friendly), smaller blood volume collections and shorter times to diagnosis. The use of this method in a larger cohort of elephants will be needed to validate whether it is applicable for diagnosing MTBC infections in elephants.

## Supporting information

**S1 Table. List of elephants and the amounts of eIFNγ detected from PBMC culture.**
(PDF)

**S2 Table. List of elephants and the amounts of eIFNγ detected from whole blood culture.**
(PDF)

**S3 Table. List of elephants and the amounts of eIFNγ detected from mitogen stimulation.**
(PDF)

**S1 File. Detection of MTBC infection by droplet digital PCR (ddPCR).**
(PDF)

## Acknowledgments

The authors are grateful to all staffs at the Elephant Kingdom Project, Zoological Park Organization (under the Royal Patronage of His Majesty the King), Warangkhana Langkaphin (DVM) at the National Elephant Institute, Forest Industry Organization, Thailand, and Panida Muanghong (DVM) from the Thai Elephant Alliance, Thailand, for the collection of elephant samples.

## Author Contributions

**Conceptualization:** Wandee Yindeeyoungyeon, Tanapat Palaga.

**Data curation:** Chitsuda Pongma, Songkiat Songthammanuphap, Wanlaya Tipkantha, Choenkwan Pabutta, Wandee Yindeeyoungyeon.

**Formal analysis:** Songkiat Songthammanuphap, Therdsak Prammananan, Saradee Warit, Wandee Yindeeyoungyeon.

**Funding acquisition:** Wandee Yindeeyoungyeon, Tanapat Palaga.

**Investigation:** Chitsuda Pongma, Therdsak Prammananan, Choenkwan Pabutta, Wandee Yindeeyoungyeon, Tanapat Palaga.

**Methodology:** Chitsuda Pongma, Songkiat Songthammanuphap, Songchan Puthong, Anumart Buakeaw, Wanlaya Tipkantha, Erngsiri Kaewkhunjob, Waleemas Jairak, Piyaporn Kongmakee, Choenkwan Pabutta, Supaphen Sripiboon, Wandee Yindeeyoungyeon, Tanapat Palaga.

**Project administration:** Wandee Yindeeyoungyeon, Tanapat Palaga.

**Resources:** Saradee Warit, Wanlaya Tipkantha, Erngsiri Kaewkhunjob, Waleemas Jairak, Piyaporn Kongmakee, Choenkwan Pabutta, Supaphen Sripiboon.

**Supervision:** Wandee Yindeeyoungyeon, Tanapat Palaga.

**Validation:** Wandee Yindeeyoungyeon.

**Writing – original draft:** Chitsuda Pongma, Wandee Yindeeyoungyeon, Tanapat Palaga.

**Writing – review & editing:** Tanapat Palaga.

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
