## [Decision Letter · Decision Letter 0]

20 Jun 2023

Using Whole Blood Cultures in Interferon Gamma Release Assays to Detect Mycobacterium tuberculosis Complex Infection in Asian Elephants (Elephas maximus)

PONE-D-23-08890

Dear Dr. anapat Palaga,

We’re pleased to inform you that your manuscript has been judged scientifically suitable for publication and will be formally accepted for publication once it meets all outstanding technical requirements.

Kind regards,

Harish Chandra, PhD

Academic Editor

PLOS ONE

“WY was supported by the National Science and Technology Development Agency (NSTDA: Project ID P-18-52114). CP is supported by a Science Achievement Scholarship of Thailand (SAST). TP is supported by the National Research Council of Thailand.”

Please respond by return e-mail so that we can amend your financial disclosure and competing interests on your behalf.

3. We note that Figure 3 in your submission contain copyrighted images. All PLOS content is published under the Creative Commons Attribution License (CC BY 4.0), which means that the manuscript, images, and Supporting Information files will be freely available online, and any third party is permitted to access, download, copy, distribute, and use these materials in any way, even commercially, with proper attribution. For more information, see our copyright guidelines: http://journals.plos.org/plosone/s/licenses-and-copyright.

a. You may seek permission from the original copyright holder of Figure 3 to publish the content specifically under the CC BY 4.0 license.

b.If you are unable to obtain permission from the original copyright holder to publish these figures under the CC BY 4.0 license or if the copyright holder’s requirements are incompatible with the CC BY 4.0 license, please either i) remove the figure or ii) supply a replacement figure that complies with the CC BY 4.0 license. Please check copyright information on all replacement figures and update the figure caption with source information. If applicable, please specify in the figure caption text when a figure is similar but not identical to the original image and is therefore for illustrative purposes only.

Additional Editor Comments (optional):

Using Whole Blood Cultures in Interferon Gamma Release Assays to Detect Mycobacterium Tuberculosis Complex Infection in Asian Elephants (Elephas maximus)" By Tanapat Palaga is well-written and scientifically sound. Further, the manuscript meets our publication criteria of ‘experiments, statistics and other analyses are performed well to a high technical standard and are described in sufficient detail’ and ‘conclusions are presented in an appropriate fashion that is supported by the data’. As an editor and reviewer, I feel the manuscript is fit for publication.

Reviewers' comments:

Reviewer's Responses to Questions

**Comments to the Author**

1. Is the manuscript technically sound, and do the data support the conclusions?

Reviewer #1: Yes

2. Has the statistical analysis been performed appropriately and rigorously? 

Reviewer #1: Yes

3. Have the authors made all data underlying the findings in their manuscript fully available?

Reviewer #1: Yes

4. Is the manuscript presented in an intelligible fashion and written in standard English?

Reviewer #1: Yes

5. Review Comments to the Author

Reviewer #1: It's a very interesting study, because it's a very unknow theme for a doctors. The new technological ways used with the IGRAS and the use of a whool blodd culture seems a very interestin procedure. Wood and beautifool work.

6. PLOS authors have the option to publish the peer review history of their article (what does this mean?). If published, this will include your full peer review and any attached files.

Reviewer #1: No

---

## [Editor Report · Acceptance letter]

18 Jul 2023

PONE-D-23-08890 

Using Whole Blood Cultures in Interferon Gamma Release Assays to Detect *Mycobacterium tuberculosis* Complex Infection in Asian Elephants (*Elephas maximus*) 

Dear Dr. Palaga:

I'm pleased to inform you that your manuscript has been deemed suitable for publication in PLOS ONE. Congratulations! Your manuscript is now with our production department. 

Kind regards, 

on behalf of

Dr. Harish Chandra 

Academic Editor

PLOS ONE